# Targeting Immunosuppressive Adenosine Signaling: A Review of Potential Immunotherapy Combination Strategies

**DOI:** 10.3390/ijms24108871

**Published:** 2023-05-17

**Authors:** David Zahavi, James W. Hodge

**Affiliations:** Center for Immuno-Oncology (CIO), Center for Cancer Research, National Cancer Institute, National Institutes of Health, Bldg. 10, Rm 8B13, 9000 Rockville Pike, Bethesda, MD 20879, USA; david.zahavi@nih.gov

**Keywords:** adenosine, immunotherapy, treatment combination, clinical trial

## Abstract

The tumor microenvironment regulates many aspects of cancer progression and anti-tumor immunity. Cancer cells employ a variety of immunosuppressive mechanisms to dampen immune cell function in the tumor microenvironment. While immunotherapies that target these mechanisms, such as immune checkpoint blockade, have had notable clinical success, resistance is common, and there is an urgent need to identify additional targets. Extracellular adenosine, a metabolite of ATP, is found at high levels in the tumor microenvironment and has potent immunosuppressive properties. Targeting members of the adenosine signaling pathway represents a promising immunotherapeutic modality that can potentially synergize with conventional anti-cancer treatment strategies. In this review, we discuss the role of adenosine in cancer, present preclinical and clinical data on the efficacy adenosine pathway inhibition, and discuss possible combinatorial approaches.

## 1. Introduction

Cancer is a complex disease that involves an intricate interplay between cancer cell-intrinsic and cell-extrinsic processes in its development and progression. Significant research advancements have identified the dysregulation of energy metabolism and immune cells as major common features of cancer cell proliferation [1]. The progress in the field since the turn of the millennium led Hanahan and Weinberg to update their hallmarks of cancer to include these areas in 2011. They further established that these characteristics contribute to the acquisition of a tumor microenvironment (TME) that is critical for cancer growth and immune evasion. Most cancer cells express antigens that can be recognized by the host immune system [2]. The remarkable success of cancer immunotherapies that activate a host anti-tumor response, such as immune checkpoint blockade or adoptive cell therapies, is revolutionizing cancer treatment paradigms. However, despite the great potential of immunotherapy, response rates for solid tumors remain low, and methods to increase the effectiveness of immunotherapy are a subject of intense investigation [3]. A growing body of evidence supports the observation that the formation of an immunosuppressive TME plays a key role in regulating anti-tumor immune responses and can limit immunotherapy efficacy [4,5]. Recent findings have further demonstrated how oncogenic pathways form an immunosuppressive milieu in the TME through a vast array of local alterations that impair immune cell infiltration and instead favor the accrual and activation of regulatory cells. Understanding these TME interactions will be crucial for the discovery of novel biomarkers and therapeutic targets and represents a promising avenue for improving immunotherapy efficacy.

Study of the tumor microenvironment has led to the identification of several metabolic pathways that are fundamental for cancer cell survival as well as metabolic interactions that play a central role in the development of resistance to immunotherapy. One such metabolite, adenosine, and its signaling pathways have been highlighted as a major regulator of anti-tumor immunity in the TME. Extracellular adenosine, which is usually present at low levels in physiologic conditions, is found at high concentrations in the TME and directly regulates cancer cell processes such as proliferation, survival, and metastasis [6]. Treatment strategies targeting the TME and adenosine specifically have emerged as promising approaches for enhancing immunotherapy [7,8]. We review the mechanisms underlying adenosine-mediated immunosuppression in the TME and its impact on immunotherapy resistance. We also summarize the preclinical data supporting the therapeutic targeting of adenosine signaling and provide an updated report on the ongoing clinical trials of adenosine-pathway-targeting agents. Finally, we discuss the potential for combining adenosine inhibition with various immunotherapeutics to optimize their effectiveness and reduce resistance as well as the remaining work to be done and future directions.

## 2. Adenosine-Pathway-Mediated Immunosuppression

Adenosine triphosphate (ATP) is the universal cellular energy currency and is present throughout the extracellular environment. Under normal physiological conditions, the extracellular levels of ATP are tightly regulated to remain in the nanomolar range. However, during inflammation, ischemia/hypoxia, tissue injury, or tumorigenesis extracellular ATP levels can increase to the micromolar range due to its release from inflammatory, apoptotic, or necrotic cells [9]. Extracellular ATP signals through P2 purinergic receptors (P2Rs) that are broadly expressed by both immune and non-immune cells and drives multiple physiological and pathological processes [10]. The current model of purinergic signaling effects on the immune response describes a delicate balance between the pro-inflammatory “danger-signal” activity of ATP and the anti-inflammatory, cytoprotective, and immunosuppressive functions of extracellular adenosine (eADO) [11]. ATP is a major regulator of inflammatory signaling and can act as a damage-associated molecular pattern (DAMP). As a DAMP, ATP can activate the NLRP3 inflammasome in cells by binding purinergic P2X7 receptors [12]. Inflammasomes are large protein complexes that play a crucial role in the innate immune system by sensing pathogenic or exogenous danger signals and triggering an inflammatory response through downstream caspase signaling cascades [13]. Of the canonical inflammasomes, NLRP3 has been found to play a key role in the TME in mediating tumor-promoting inflammation, immune evasion, proliferative signaling, invasion, angiogenesis, and inhibition of apoptosis [14,15]. Additional ATP released from stressed and/or dying cells can also induce immunogenic cell death and is important for immune cell recognition of active infection, while accumulation of adenosine aims to restore tissue homeostasis and prevent an excessive inflammatory reaction. Like ATP, eADO in normal tissues is held constant around nanomolar concentrations but can increase over 100-fold in the TME [16]. Under normal physiological conditions, adenosine is mainly found intracellularly and is generated during the conversion of S-adenosylhomocysteine to adenosine and homocysteine by Adenosylhomocysteinase [17]. The amount of adenosine in the extracellular space is controlled by equilibrative nucleoside transporters (ENTs), which shuttle adenosine based on its concentration gradient, and concentrative nucleoside transporters (CNTs), which allow the intracellular influx of adenosine against its concentration gradient. In healthy tissues, adenosine regulates numerous physiological processes but mainly acts to maintain homeostasis following injury by restricting immune responses and promoting wound healing [18]. In contrast, the levels of eADO in the TME are regulated by multiple systems of enzymes and transporters that produce, degrade, and recycle purine metabolites (Figure 1). Extracellular adenosine in the TME can be produced by simple diffusion or active transport of intracellular adenosine but is mainly generated through two pathways. In the first, ATP in the extracellular environment is hydrolyzed to ADP and AMP, and then AMP is converted to eADO by the ectonucleotidases CD39 and CD73 [19]. Therefore, eADO can rapidly accumulate in the TME due to the higher levels of ATP present and the overexpression of CD39 and CD73 found on multiple cell types in the TME [20]. In the second pathway, AMP is produced from NAD+ by the enzymatic actions of CD38 and CD203a, and is then converted to eADO by CD73 [21]. Furthermore, cancer cells commonly have genetic mutations that promote altered purine metabolism to facilitate increased production or reduced degradation of eADO [22]. Extracellular adenosine is removed from the environment by cell surface ADO deaminase, which converts it to inosine, and by nucleoside transporters that bring eADO intracellularly for conversion back to AMP by ADO kinases [23]. Hypoxic conditions, such as those found in the TME, can inhibit these eADO-consuming pathways, which further amplifies the increase in adenosine signaling [24].

The main signaling actions of eADO are mediated by four G-protein-coupled receptors classified as the subtypes A1, A2_A_, A2_B_, and A3. The A1, A2_A_, and A3 subtypes are considered high-affinity, and A2_B_ by contrast is a low-affinity receptor that is only active in pathological conditions when eADO levels are elevated [25]. Adenosine receptors are widely distributed in many different tissues including the nervous, cardiovascular, gastrointestinal, and immune systems due to adenosine’s role as a ubiquitous extracellular signaling molecule [26]. In addition to regulating metabolic homeostasis in most cells by manipulating adenosine levels, adenosine receptors play important roles in modulating cardiac contractility and vasodilation, synaptic transmission in the brain and sleep cycles, and inflammatory responses [27]. The A1 and A3 subtypes are coupled to G_i/o_ and inhibit the activity of adenylate cyclase, whereas A2_A_ and A2_B_ receptors are G_s_ family members and activate adenylate cyclase, thus triggering cAMP-dependent downstream signaling events. Additionally, adenosine receptors can signal through cAMP-independent pathways. The A2_B_ and A3 receptors can trigger phospholipase C-mediated pathway activation via G_q/11_ proteins and A2_A_ and A2_B_ receptors have been found to activate ERK, p38 MAPK, and/or PI3K–AKT–mTOR-dependent pathways in various cell types [28]. In an oncogenic setting, the immunosuppressive effects of eADO are mainly induced via binding by the A2_A_ and A2_B_ receptors [29]. These two receptor subtypes are present on a variety of cell types located in the TME, with A2_A_ receptors ubiquitously expressed by immune cells and A2_B_ receptors predominantly found on myeloid cells. The adenosine receptor system can exert a variety of effects on the differentiation, maturation, and activation state of cells in the TME depending on cell type (Figure 2). All four adenosine receptors have been reported to be expressed by cancer cells in both hematological malignancies and solid tumors [30]. Adenosine signaling has been found to play a role in regulating cancer cell apoptosis, proliferation, and metastasis with A1, A2_A_, and A2_B_ receptor activation associated with pro-tumorigenic effects and A3 receptor signaling exerting anti-tumor effects [31]. Additionally, A2_A_ and A2_B_ receptors are often overexpressed by cancer cells and stimulation can enhance tumor growth [32,33]. Vascular endothelial cells and pericytes in the TME also express A2_A_ and A2_B_ receptors, through which eADO signaling can initiate and support tumor angiogenesis [34,35]. Furthermore, activation of A2_A_ and A2_B_ receptors expressed on a range of myeloid cells including monocytes, macrophages, and dendritic cells leads to differentiation into pro-tumorigenic subtypes and increased release of pro-tumorigenic and anti-immunity cytokines that promote an overall immunosuppressive TME [36,37]. In the past decade, a substantive body of work has demonstrated that the primary mechanism for eADO subversion of anti-tumor immunity is through A2_A_ and/or A2_B_ signaling in T and natural killer (NK) cells. When T cells become activated, they increase their expression of adenosine receptors on their cell surface and most of the suppressive signaling is dependent on the predominantly expressed A2_A_ receptor subtype [38]. Stimulation of CD8+ T-cell A2_A_ receptors results in reduced secretion of key immune effector molecules such as IL-2, TNFα, and interferon (IFN)-γ and promotes T-cell anergy. Additionally, A2_A_ receptor signaling in naive CD4+ T cells has been observed to drive their differentiation toward the immunosuppressive CD4+FOXP3+ T-regulatory (Treg) phenotype and enhances their expression of CD39 and CD73 [39]. Of all the lymphocyte populations, NK cells express the highest levels of A2_A_ receptors, and this is considered the main mechanism for eADO-mediated immunosuppression similar to T cells [40]. Crucially, eADO-A2_A_ receptor signaling in NK cells directly suppresses their maturation in the TME and leads to immunometabolic reprogramming that results in decreased proliferation, survival, and cytotoxic function [41,42]. Collectively, these studies suggest that eADO plays a key role in modulating the anti-tumor immune response by driving the formation of an immunosuppressive TME and is an attractive potential target for therapeutic intervention.

## 3. Preclinical and Clinical Data Supporting Adenosine-Pathway-Targeted Therapies

Cancer immunotherapies such as immune checkpoint blockade (ICB) have revolutionized cancer treatment and have renewed interest in therapies that can restore anti-tumor immunity. However, despite the notable successes of ICB, only a minority of solid tumor patients have durable responses, which has highlighted the need for further research in understanding immune evasion mechanisms to identify novel therapeutic targets [43]. As described in the previous sections, the adenosine signaling pathway has been shown to regulate immune cell responses in the TME and is an attractive target for immunotherapeutic development. There is a wealth of both preclinical and clinical data that support further interrogation of adenosine signaling inhibition for numerous solid tumors. Adenosine signaling pathway components such as CD39, CD73, and A2_A_ receptors are overexpressed by multiple cell types in the TME of various cancers, including colorectal, gastric, head and neck, breast, and brain cancer, and often correlate with an immunosuppressive signature, aggressiveness, and poor prognosis in patients [44,45,46,47,48]. In 1975, Chu et al. became the first to demonstrate that eADO could suppress T-cell activity against cancer cells in vitro [49]. Several landmark studies subsequently established the primary role of A2_A_ receptors in mediating immunosuppressive signaling and that targeting adenosine receptors could potently enhance anti-tumor responses both in vitro and in vivo [50,51,52]. Additional preclinical evidence has shown that targeting the production of eADO in the TME via CD39 and CD73 is another promising strategy for restoring anti-tumor immunity. Reducing CD73 expression in tumor cell lines sensitized them to T-cell-mediated killing, and anti-CD73 antibodies could reduce tumor growth and metastasis by activating NK and T-cell responses [53,54]. Likewise, CD39-targeting monoclonal antibodies have been shown to inhibit eADO production in the TME, effectively suppress metastasis, upregulate the expression of activation markers and cytotoxic granule components in immune cells, and increase the anti-tumor functions of infiltrating NK and T cells [55,56,57]. Monoclonal antibodies and small-molecule inhibitors that target CD39, CD73, and adenosine receptors are all under development for cancer therapy. Compared to anti-CD39 and anti-CD73 antibodies, small-molecule inhibitors may have better penetration into solid tumors and become more widely available in the TME [58]. Given the accumulating preclinical research indicating the benefit of targeting eADO to increase anti-tumor immunity, there has been accelerated initiation of clinical trials.

Early results from clinical trials of eADO-pathway-targeting agents have demonstrated good tolerability and some efficacy as a monotherapy. One of the first drugs developed to target adenosine-mediated immunosuppression was the dual A2_A_/A2_B_ receptor antagonist AB928 or etrumadenant, which was found to be safe, orally bioavailable, and possessed immunomodulatory activity in healthy volunteers [59]. Etrumadenant is now being further evaluated in multiple phase I/II studies for its anti-cancer efficacy [60]. The first clinical report confirming the utility of adenosine pathway antagonism for cancer treatment came from a phase I trial of CPI-444 or ciforadenant, a small-molecule inhibitor with selectivity for A2_A_ receptors, which demonstrated monotherapy activity in renal cell carcinoma patients, even those who were treatment-refractory to anti-PD-L1 therapy [61]. Phase I studies of additional A2_A_ receptor antagonists further established these drugs as well tolerated with minimal side effects and capable of inducing clinical responses as both a monotherapy and in combination with other immunotherapies [62,63]. Several anti-CD39 and anti-CD73 monoclonal antibodies have also begun entering clinical trials. Recently, results from the first-in-human study of the anti-CD73 antibody oleclumab revealed a manageable safety profile and some evidence of anti-tumor activity [64]. There are comparatively few clinical studies of anti-CD39 antibodies; however, at least three (TTX-30, SRF617, and IPH5201A) have trials ongoing [65]. Unfortunately, there are no current clinical-grade CD39 inhibitors, but a few small-molecule inhibitors of CD73 have entered phase I clinical trials. The CD73-selective inhibitors LY3475070 and AB-680 are currently being evaluated with results forthcoming. There are at least 54 active clinical trials for eADO-pathway-targeting agents (Table 1). Overall, the available clinical data for adenosine inhibition are very promising and support further investigation, yet several hurdles to approval remain. Although numerous adenosine receptor inhibitors have been developed and experienced success preclinically, achieving sufficient affinity and target selectivity has proved challenging in patients. A comprehensive review of the drug development considerations for adenosine-signaling-targeted therapies indicates further optimization of the physicochemical and pharmacokinetic properties is needed for clinical efficacy [66,67]. Moreover, recent findings indicate that eADO pathway inhibition alone may not be sufficient to restore anti-tumor immunity and further study of alternative mechanisms of action is needed [68].

## 4. Potential Combinations with Adenosine-Pathway-Targeted Therapy

Strong rationale exists for combining adenosine-pathway-targeted therapies with traditional chemotherapy or immunotherapy. Immune checkpoint blockade can activate T cells and increase their infiltration into solid tumors, but resistance is common. Adenosine signaling is a non-redundant immunosuppressive mechanism in the TME that, if inhibited, could further unleash the cytotoxic potential of infiltrating immune cells. First, there are multiple pathways that contribute to eADO-mediated immunosuppression in the TME; therefore, there may be synergy when combining eADO-generating pathway inhibition with eADO receptor antagonism. A study of dual therapy with an anti-CD73 mAb and an A2_A_ receptor antagonist found there was superior anti-tumor immunity with the combination compared to either treatment alone [69]. Early reports from a clinical trial investigating a combination of A2_A_ receptors and CD73 inhibition similarly found an increase in response rate without a marked increase in the adverse event rate compared to either monotherapy [70]. Another potential strategy for enhancing efficacy is the co-inhibition of CD39 and CD73, and this is supported by preclinical data indicating compensatory mechanisms when either is targeted alone [71]. Moreover, CD39 and CD73 have pro-tumorigenic functions beyond their enzyme activity, including potentiating tumor cell adhesion, migration, and metastasis [72]. For example, an antibody targeting CD73 that did not affect eADO production could induce internalization of CD73, which inhibited metastasis formation [73]. Thus, combining adenosine-pathway-targeted agents with other therapies may have further additive or synergistic effects.

### 4.1. Combination with Chemotherapy and Radiotherapy

Both radiotherapy approaches as well as several chemotherapeutic agents have shown the capacity to induce ATP release, elevate the expression of CD39 and CD73, and stimulate eADO production in the TME [74,75]. Chemotherapy remains widely used for many cancer patients, and several regimens have been established as immunomodulatory with efficacy dependent on an anti-tumor immune response. Thus, there is potential synergy when combining adenosine pathway inhibition with chemotherapy. Importantly, evidence suggests that chemotherapy can directly upregulate adenosine signaling. For example, doxorubicin was shown to induce CD73 expression on cancer cells and increase eADO levels [76]. Moreover, in a preclinical model of colon cancer, inhibition of CD73 increased the therapeutic response to immunogenic chemotherapy [77]. Similarly, antagonism of A2_A_ and A2_B_ receptors augmented the efficacy of chemotherapy and led to greater immune activation, reduced tumor growth, and prolonged survival [78]. Clinical trials evaluating these combinations, such as the phase II SYNERGY trial (NCT03616886) of the anti-CD73 antibody oleclumab in combination with the anti-PD-L1 antibody durvalumab and/or chemotherapy in patients with triple-negative breast cancer, are now underway. Likewise, an anti-CD73 monoclonal enhanced the efficacy of radiotherapy by increasing the presence of CD8+ T cells while decreasing the number of Tregs within irradiated tumors [79]. Interestingly, irradiated tumor cells upregulated their expression of CD73 and CD38 but not CD39, suggesting utilization of the non-canonical pathway of eADO generation [80]. A phase II clinical trial to assess the addition of oleclumab to radiotherapy in luminal B breast cancer patients has been initiated (NCT03875573).

### 4.2. Combination with Immune Checkpoint Inhibitors

Adenosine signaling has been identified as a critical regulator of the immune response in the TME and is a prime candidate for combination with other immunotherapies. This is underscored by evidence that CD73+ tumor cells are more resistant to anti-PD-1 ICB and that anti-PD-1 antibodies increase adenosine receptor expression on tumor-infiltrating CD8+ T cells [81]. Interestingly, the reverse is also true, and antibodies targeting adenosine receptors have been observed to increase PD-1 expression by CD8+ T cells [82]. Evidence of these reciprocal interactions between adenosine receptor and PD-1/PD-L1 expression suggests combined inhibition will be superior to single-agent treatments. Accordingly, co-administration of an A2_A_ receptor inhibitor and an anti-PD1 antibody resulted in significantly lower metastatic burden and increased survival in a preclinical model [83]. Moreover, concomitant blockade of CD73 and PD-1 has been found to be more effective than either monotherapy alone in mouse models [84]. A multitude of studies have further demonstrated that anti-CD39 and anti-CD73 antibodies have synergy with various immune checkpoint inhibitors [85]. Increased adenosine production due to upregulation of CD38 by tumor cells was identified as another mechanism of resistance to PD-1 and PD-L1 checkpoint blockade, and inhibition of CD38 was shown to significantly improve responses to an anti-PD-L1 antibody [86].

### 4.3. Combination with Cellular Therapies

Targeting the adenosinergic pathway can also be combined with adoptive cell immunotherapy. Chimeric antigen receptor T cells (CAR-T) upregulate the expression of the A2_A_ receptor upon activation, a mechanism that might contribute to treatment resistance. CRISPR-mediated deletion of the A2_A_ receptor in CAR-T cells enhanced their efficacy in both in vitro and in vivo models, especially when combined with PD-1 inhibition [87]. A preclinical study of the dual A2_A_ and A2_B_ receptor antagonist etrumadenant found that treatment enhanced CAR-T-cell cytokine secretion and proliferation and augmented cytolysis of cancer cells in vitro and in vivo [88]. Recent insights into NK cell function have revealed that the adenosine pathway is a powerful regulator of NK cell activation, maturation, and function, suggesting that targeting adenosinergic signaling may be a promising avenue for improving NK-cell-based therapies [89]. However, there has been limited research on the effects of adenosine receptor targeting on NK cells. In one report, a combinatorial blockade of the immune checkpoint TIGIT and either CD39 or the A2_A_ receptor synergistically strengthened NK-92 cell-mediated cytotoxicity against leukemia cells [90]. Clinical trials of eADO pathway inhibitors in combination with cellular immunotherapies have yet to be initiated, but further study is warranted.

## 5. Conclusions

Escaping the host anti-tumor immune response is one of the hallmarks of cancer. Tumors employ a variety of immunosuppressive mechanisms that create an unfavorable TME for immune cells, and overcoming these mechanisms is a major focus of immunotherapy. Adenosine, a metabolite of ATP, and its signaling pathway have been shown to have broad immunosuppressive functions. In the TME, high levels of adenosine are primarily produced through the actions of the ectoenzymes CD39 and CD73. These proteins, along with the downstream A2_A_ and A2_B_ receptors found on immune cells, are promising targets for enhancing anti-tumor immunity. Clinical-grade drugs targeting this pathway have entered human trials both alone and in combination with other immunotherapeutic approaches. Preliminary reports from these trials indicate that adenosine-targeted therapies have a clinically relevant mechanism of action, favorable safety profile, and moderate efficacy, even in patients who are refractory to ICB. However, several challenges remain in the development of adenosine pathway inhibitors. Firstly, there is ubiquitous expression of adenosine receptors by many cell types; thus, the potential for off-target side effects exists. The development of novel compounds must ensure high levels of selectivity. Secondly, some of the underlying mechanisms of these therapies remain relatively unknown, particularly in distinct immune cell subtypes. Future studies should focus on elucidating the function of eADO in different components of the TME. Finally, while adenosine pathway antagonists have activity in multiple indications, predictive biomarkers are needed to identify patients most likely to benefit from these therapies. In conclusion, eADO is a powerful mediator of immunosuppression and inhibitors of this pathway are a promising therapeutical approach that could potentially augment conventional anti-cancer treatments.

## Figures and Tables

**Figure 1 ijms-24-08871-f001:**
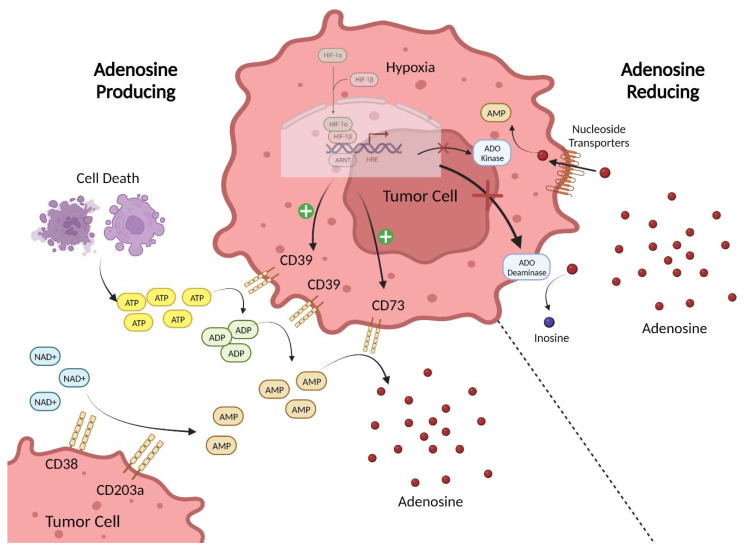
Sources of Adenosine in the TME. The major adenosine (eADO)-producing pathway in the TME involves the processing of the precursor ATP by ectonucleotidases. ATP is released into the extracellular environment due to cell death. ATP accumulating in the extracellular milieu is then enzymatically converted via the canonical pathway involving the sequential hydrolysis of ATP to ADP and AMP by CD39 and subsequently the hydrolysis of AMP to eADO by CD73. In the non-canonical pathway, precursor NAD+ substrate is acted on by CD38 to generate ADP-ribose (ADPR) that can be converted to AMP by CD203a, which is then hydrolyzed by CD73 to produce eADO. To reduce eADO levels in the extracellular space, adenosine can be re-uptaken into the cell via nucleoside transporters, where it is converted to AMP by ADO kinase or converted to inosine by cell surface ADO deaminase. Hypoxic signaling in tumor cells promotes expression of CD39 and CD73 while simultaneously inhibiting ADO kinase and ADO deaminase activity. Created with BioRender.com.

**Figure 2 ijms-24-08871-f002:**
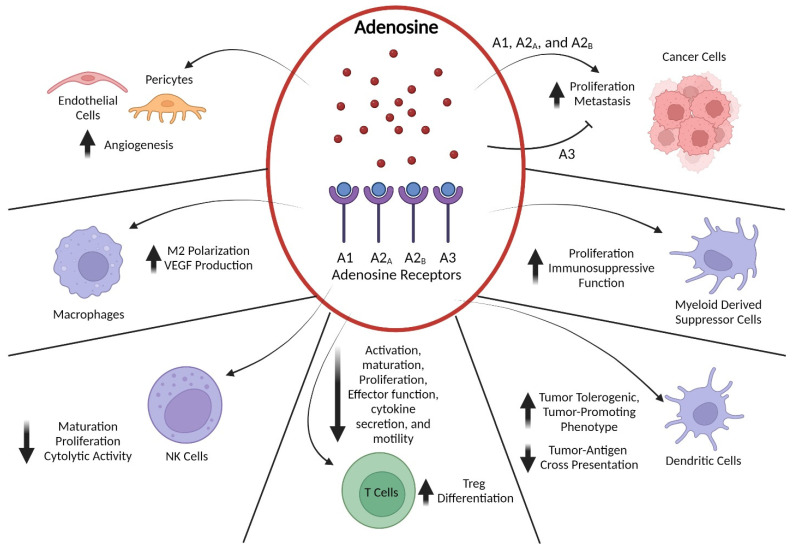
Adenosine Signaling Effects on Cells in the TME. Adenosine accumulated in the TME exerts its effects on a variety of cell types by binding to A1, A2_A_, A2_B_, and A3 receptors. Signaling through A2_A_ and A2_B_ receptors induces cyclic AMP (cAMP)-dependent pathways, whereas activation of A1 and A3 receptors inhibits cAMP generation. In cancer cells, adenosine binding to A1, A2_A_, and A2_B_ receptors stimulates proliferation, invasion, and metastasis, whereas binding to A3 receptors inhibits those processes. Both A2_A_ and A2_B_ receptor signaling exert potent immunomodulatory effects on myeloid-derived suppressor cells and enhance their immunosuppressive cytokine secretion. Similarly, A2_A_ and A2_B_ receptor signaling suppresses professional antigen-presenting cells such as dendritic cells by reducing their capacity for tumor-antigen cross-presentation through downregulation of MHC class II expression, while also promoting their production of tumor-tolerogenic and/or -promoting factors such as IL-6, IL-10, and TGFβ. In T cells, A2_A_ binding leads to a multi-layered suppression of anti-tumor immune function by dampening T-cell receptor and co-stimulatory signals, thereby blocking effector T-cell activation, maturation, proliferation, motility, and secretion of multiple cytokines. Additionally, activation of A2_A_ receptors in CD4+ T cells promotes their differentiation into regulatory T (Treg) cells and increases their immunosuppressive capabilities. Likewise, NK cells express high levels of A2_A_ receptors that, when bound by adenosine, limit their maturation, proliferation, and cytotoxic function. Activation of A2_A_ and A2_B_ receptors in macrophages triggers differentiation into the tumor-associated M2 phenotype that produces tumor-promoting cytokines such as vascular endothelial growth factor (VEGF). Finally, adenosine-mediated signaling through A2_A_ and A2_B_ receptors on endothelial cells and pericytes promotes angiogenesis and limits trafficking of immune cells. Created with BioRender.com.

**Table 1 ijms-24-08871-t001:** Active Clinical Trials for Adenosine-Targeted Therapies.

NCT Number	Therapeutic	Target	Indication	Status
eADO Production Targeted				
NCT04336098	SRF617	Anti-CD39 mAb	Advanced Solid Tumors	Phase I, Not Recruiting
NCT05717348	ES014	Anti-CD39xTGF-β bsAb	Locally Advanced/Metastatic Solid Tumors	Phase I, Recruiting
NCT05234853	PUR001	Anti-CD39 mAb	Advanced Solid Tumors	Phase I, Recruiting
NCT03884556	TTX-030	Anti-CD39 mAb	Advanced Solid Tumors	Phase I, Not Recruiting
NCT05508373	JS019	Anti-CD39 mAb	Advanced Solid Tumors	Phase I, Recruiting
NCT05075564	ES002023	Anti-CD39 mAb	Locally Advanced/Metastatic Solid Tumors	Phase I, Not Recruiting
NCT05374226	JS019	Anti-CD39 mAb	Advanced Solid Tumors and Lymphomas	Phase I, Recruiting
NCT04306900	TTX-030	Anti-CD39 mAb	Advanced Solid Tumors	Phase I, Not Recruiting
NCT05742607	IPH5201	Anti-CD39 mAb	Non-Small-Cell Lung Cancer	Phase II, Recruiting
NCT04104672	AB680	Small-Molecule CD73 Inhibitor	Gastrointestinal Cancers	Phase I, Recruiting
NCT05329766	AB680	Small-Molecule CD73 Inhibitor	Advanced Gastrointestinal Cancers	Phase II, Recruiting
NCT05227144	ORIC-533	Small-Molecule CD73 Inhibitor	Relapsed or Refractory Multiple Myeloma	Phase I, Recruiting
NCT05431270	PT199	Anti-CD73 mAb	Locally Advanced/Metastatic Solid Tumors	Phase I, Recruiting
NCT04940286	Oleclumab	Anti-CD73 mAb	Pancreatic Cancer	Phase II, Recruiting
NCT05001347	TJ004309	Anti-CD73 mAb	Locally Advanced/Metastatic Solid Tumors	Phase II, Not Recruiting
NCT05559541	AK119	Anti-CD73 mAb	Advanced Solid Tumors	Phase I, Recruiting
NCT05689853	AK119	Anti-CD73 mAb	Advanced Solid Tumors	Phase I, Recruiting
NCT05173792	AK119	Anti-CD73 mAb	Advanced Solid Tumors	Phase I, Recruiting
NCT04322006	TJ004309	Anti-CD73 mAb	Advanced Solid Tumors	Phase I, Recruiting
NCT03835949	TJ004309	Anti-CD73 mAb	Locally Advanced/Metastatic Solid Tumors	Phase I, Not Recruiting
NCT05001347	TJ004309	Anti-CD73 mAb	Ovarian Cancer/Select Solid Tumors	Phase II, Not Recruiting
NCT03454451	CPI-006	Anti-CD73 mAb	Advanced Solid Tumors	Phase I, Not Recruiting
NCT03875573	Oleclumab	Anti-CD73 mAb	Breast Cancer	Phase II, Recruiting
NCT02503774	Oleclumab	Anti-CD73 mAb	Advanced Solid Tumors	Phase I, Not Recruiting
NCT03616886	Oleclumab	Anti-CD73 mAb	Recurrent or Metastatic TNBC	Phase I/II, Not Recruiting
NCT05174585	JAB-BX102	Anti-CD73 mAb	Advanced Solid Tumors	Phase I/II, Recruiting
NCT05632328	AGEN1423	Anti-CD73xTGF-β-Trap bsAb	Pancreatic Cancer	Phase II, Not Recruiting
NCT05246995	IBI325	Anti-CD73 mAb	Advanced Solid Tumors	Phase I, Not Recruiting
NCT05119998	IBI325	Anti-CD73 mAb	Advanced Solid Tumors	Phase I, Not Recruiting
NCT05143970	IPH5301	Anti-CD73 mAb	Advanced Solid Tumors	Phase I, Recruiting
NCT05173792	AK119	Anti-CD73 mAb	Advanced Solid Tumors	Phase I, Recruiting
NCT04572152	AK119	Anti-CD73 mAb	Advanced Solid Tumors	Phase I, Not Recruiting
NCT04672434	Sym024	Anti-CD73 mAb	Advanced Solid Tumors	Phase I, Recruiting
NCT04237649	NZV930	Anti-CD73 mAb	Advanced Solid Tumors	Phase I, Recruiting
NCT04668300	Oleclumab	Anti-CD73 mAb	Advanced Sarcoma	Phase II, Recruiting
Adenosine Receptor Targeted				
NCT04976660	TT-4	Small-Molecule A2_B_ Receptor Inhibitor	Advanced Solid Tumors	Phase I/II, Not Recruiting
NCT04969315	TT-10	Small-Molecule A2_A_ Receptor Inhibitor	Advanced Solid Tumors	Phase I/II, Not Recruiting
NCT05198349	M1069	Small-Molecule Dual A2_A_/A2_B_ Receptor Inhibitor	Advanced Solid Tumors	Phase I, Not Recruiting
NCT05272709	TT-702	Small-Molecule A2_B_ Receptor Inhibitor	Advanced Solid Tumors	Phase I/II, Recruiting
NCT04660812	Etrumadenant	Small-Molecule Dual A2_A_/A2_B_ Receptor Inhibitor	Metastatic Colorectal Cancer	Phase I/II, Not Recruiting
NCT05501054	Ciforadenant	Small-Molecule A2_B_ Receptor Inhibitor	Renal Cell Carcinoma	Phase I/II, Recruiting
NCT05060432	Inupadenant	Small-Molecule A2_A_ Receptor Inhibitor	Advanced Solid Tumors	Phase I/II, Recruiting
NCT05117177	Inupadenant	Small-Molecule A2_A_ Receptor Inhibitor	Advanced Solid Tumors	Phase I, Recruiting
NCT02740985	AZD4635	Small-Molecule A2_A_ Receptor Inhibitor	Advanced Solid Tumors	Phase I, Not Recruiting
NCT04089553	AZD4635	Small-Molecule A2_A_ Receptor Inhibitor	Prostate Cancer	Phase II, Not Recruiting
NCT05234307	PBF-1129	Small-Molecule A2_B_ Receptor Inhibitor	Recurrent or Metastatic Non-Small-Cell Lung Cancer	Phase I, Recruiting
NCT03274479	PBF-1129	Small-Molecule A2_B_ Receptor Inhibitor	Non-Small-Cell Lung Cancer	Phase I, Not Recruiting
NCT04381832	Etrumadenant	Small-Molecule Dual A2_A_/A2_B_ Receptor Inhibitor	Metastatic Prostate Cancer	Phase I/II, Recruiting
NCT04892875	Etrumadenant	Small-Molecule Dual A2_A_/A2_B_ Receptor Inhibitor	Advanced HNSCC	Phase I, Not Recruiting
NCT05024097	Etrumadenant	Small-Molecule Dual A2_A_/A2_B_ Receptor Inhibitor	Rectal Cancer	Phase II, Recruiting
NCT04580485	INCB106385	Small-Molecule Dual A2_A_/A2_B_ Receptor Inhibitor	Advanced Solid Tumors	Phase I, Recruiting
Combined Pathway Targeted				
NCT03381274	Oleclumab and AZD4635	Anti-CD73 mAb and Small-Molecule A2_A_ Receptor Inhibitor	Epidermal Growth Factor Receptor Mutant Non-Small-Cell Lung Cancer	Phase I/II, Not Recruiting
NCT03454451	CPI-006 and Ciforadenant	Anti-CD73 mAb and Small-Molecule A2_A_ Receptor Inhibitor	Advanced Solid Tumors	Phase I, Not Recruiting
NCT04989387	INCA00186 and INCB106385	Anti-CD73 mAb and Small-Molecule Dual A2_A_/A2_B_ Receptor Inhibitor	Advanced Solid Tumors	Phase I, Recruiting

mAb, monoclonal antibody; bsAb, bi-specific antibody; TNBC, triple-negative breast cancer, HNSCC, head and neck squamous cell carcinoma.

## Data Availability

Not applicable.

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
