# Peer review of "Targeting Immunosuppressive Adenosine Signaling: A Review of Potential Immunotherapy Combination Strategies"

_ijms, 2023, doi:10.3390/ijms24108871_

Round 1

Reviewer 1 Report

The Review describes the role of adenosine signaling in tumor immunity. It is an interesting manuscript, however, there are several issues that should be addressed:

1. Authors should add the role of ATP in the stimulation of inflammasome 

2. Many sentences throughout the manuscript are missing references

3. All figures are missing explanation and the references under the figures 

Author Response

REVIEWER 1:

  1. Authors should add the role of ATP in the stimulation of inflammasome 

Authors’ Response: We thank the reviewer for the opportunity to expand on this. We have added a paragraph detailing the role of ATP in the stimulation of the inflammasome and its relevance to the TME (Page 3, Lines 102-109).

  1. Many sentences throughout the manuscript are missing references.

Authors’ Response: We have reviewed the manuscript to ensure all references are included.

  1. All figures are missing explanation and the references under the figures 

Authors’ Response: We appreciate this comment. As such we have now added a description of each figure.

Reviewer 2 Report

This review article is well-written and provides detailed information about the immunosuppressive adenosine signaling pathways in the tumor immune microenvironment. However, some improvements can be made to enhance its overall impact.

In the initial section titled "Adenosine-mediated immunosuppression," the provided information is commonly found in numerous other review articles. To make this review article more interesting and unique, it is important to discuss every aspect of the Adenosine pathway in detail. This includes describing how adenosine is generated and its role under normal physiological conditions.

Additionally, each adenosine receptor should be discussed under a separate heading, highlighting the signaling pathways they induce and their respective physiological roles.

The section on 'Potential combination with Adenosine pathway-targeted therapy' can be further elaborated by using subheadings to discuss various combinations, such as 'potential combination with immune checkpoint inhibitors,' 'potential combination with CAR-T cells,' 'potential combination with chemotherapy,' and other novel combinations that the author believes can be utilized.

By incorporating these suggestions, the review article will provide a comprehensive and engaging overview of the immunosuppressive adenosine signaling pathways in the tumor immune microenvironment.

Author Response

REVIEWER 2:

  1. In the initial section titled "Adenosine-mediated immunosuppression," the provided information is commonly found in numerous other review articles. To make this review article more interesting and unique, it is important to discuss every aspect of the Adenosine pathway in detail. This includes describing how adenosine is generated and its role under normal physiological conditions.

Authors’ Response: We agree with the reviewer and as such we have now added a discussion of how adenosine is generated and its role in normal physiological conditions (Page 3, Lines 114-122).

  1. Additionally, each adenosine receptor should be discussed under a separate heading, highlighting the signaling pathways they induce and their respective physiological roles

Authors’ Response: We appreciate this comment. We have added information highlighting the physiological roles of adenosine receptors and the diverse signaling pathways they can induce (Page 3, Lines 143-148) and (Page 3-4, Lines 151-155). Our review is focused on the role of adenosine receptors in cancer and have kept the discussion of adenosine receptors and signaling centered on the effects on cell types in the TME.

  1. The section on 'Potential combination with Adenosine pathway-targeted therapy' can be further elaborated by using subheadings to discuss various combinations, such as 'potential combination with immune checkpoint inhibitors,' 'potential combination with CAR-T cells,' 'potential combination with chemotherapy,' and other novel combinations that the author believes can be utilized.

Authors’ Response: We agree with the reviewer and as such we have divided the section on potential combinatorial approaches to include subheading for discussing various combinations. We have also added additional information to expand this section (Page 10-11, Lines 327-337, 349-354, and 361-372).

Round 2

Reviewer 1 Report

The authors have addressed the comments 

Reviewer 2 Report

Authors satisfactorily answered all my comments.